# Projection-type see-through holographic three-dimensional display

Koki Wakunami[1], Po-Yuan Hsieh[2], Ryutaro Oi[1], Takanori Senoh[1], Hisayuki Sasaki[1], Yasuyuki Ichihashi[1], Makoto Okui[1], Yi-Pai Huang[2] & Kenji Yamamoto[1]

Owing to the limited spatio-temporal resolution of display devices, dynamic holographic three-dimensional displays suffer from a critical trade-off between the display size and the visual angle. Here we show a projection-type holographic three-dimensional display, in which a digitally designed holographic optical element and a digital holographic projection technique are combined to increase both factors at the same time. In the experiment, the enlarged holographic image, which is twice as large as the original display device, projected on the screen of the digitally designed holographic optical element was concentrated at the target observation area so as to increase the visual angle, which is six times as large as that for a general holographic display. Because the display size and the visual angle can be designed independently, the proposed system will accelerate the adoption of holographic three-dimensional displays in industrial applications, such as digital signage, in-car head-up displays, smart-glasses and head-mounted displays.

[1] National Institute of Information and Communications Technology, Tokyo 184-8795, Japan. [2] National Chiao Tung University, Hsinchu, Taiwan 30010, China. Correspondence and requests for materials should be addressed to K.W. (email: k.wakunami@nict.go.jp).

Technology that can achieve 'three-dimensional (3D) images floating in air' an important goal in display research. Since holography is the only technique that can reproduce all the depth cues in the human visual system, holography is the most promising technology for realizing 3D visualization. Static hologram print services have already been commercialized[1,2], and 3D models designed in computer graphics can be printed out as full-colour holograms based on the well-known holographic stereogram (HS) technique. In contrast, dynamic holographic displays are still at the research stage, and there are not yet any practical systems on the market. 'Mark-1' developed by a group at the Massachusetts Institute of Technology (MIT) was a pioneering technology, and it achieved 3D reconstruction covering a $25 \times 25 \times 25$ mm volume in full-colour with a viewing angle of $15°$ by employing horizontal parallax only[3]. Since then, many research groups have attempted to improve various quality factors of holographic displays via a number of approaches, such as the tiling of spatial light modulators (SLMs) to expand the display size[4] or the viewing angle[5,6], the horizontal parallax only approach to markedly reduce the cost of calculating the hologram data and increase the horizontal viewing angle[7], eye-tracking to reduce the computational cost[8], the use of updatable photosensitive materials to overcome the resolution limitation of current SLMs[9] and so forth.

From the viewpoint of the practical use of holographic 3D displays, it is important to be able to freely design the display size and the visual angle to view the entire display area to realize a wide range of applications. However, as long as display devices having severe spatio-temporal resolution limitations are used in hologram reconstruction, we still face the inherent trade-off between the display size and the maximum diffraction angle of the controllable light. A small diffraction angle results in a small visual angle, since these angles are identical when the hologram on the display device is illuminated by a plane wave. This trade-off between the display size and the visual angle has been well formulated, as the theory of the space-bandwidth product (SBP)[10].

One approach to overcoming this problem so as to increase both the display size and the visual angle at the same time is to concentrate the reconstructed light at a target observation area by using a convex lens[11] or a custom-designed mirror[12]. However, it is costly to fabricate these optical elements with a diameter large enough to cover the whole display, and in addition, the design of the display system, such as the location of the observation area, is quite limited due to the limitations of the optical characteristics of general optical elements.

In 1977, Komar[13] proposed a primary concept of holographic projection with a holographic screen as a 'holographic cinematography' technique, in which an analogue holographic image recorded on a 70 mm hologram film was projected on the holographic screen by a projector lens. The projected image was then reflected to the assumed observation point by a screen function that is a replica of a large concave mirror. In that paper, however, Komar pointed out several problems, namely, the difficulty of fabricating a large holographic screen, the severe image distortions of 3D space owing to the mismatch between the axial and lateral dimensions caused by the enlarged projection, and the image distortion due to observation of the hologram screen from an oblique direction.

Recently, digital holographic projection methods in which a two-dimensional (2D) image reproduced by holographic reconstruction of digital hologram data is projected on a diffuse screen have been proposed[14–16]. Image size zooming and focusing can be controlled during the holographic data calculation process, and this is expected to simplify and reduce the size of current 2D projectors. However, these approaches cannot realize 3D reconstruction, since the directionality of the holographic reconstructed light is lost due to the use of a diffuse screen.

A screen based on a holographic optical element (HOE) would be well suited for realizing 3D reconstruction using holographic projection because of its potential to possess arbitrary optical characteristics at multiple wavelengths for a full-colour display system. However, in common HOE fabrication methods, the optical characteristic of the HOE is once analogy shaped by using an optical element then recorded into a hologram, and thus, the achievable optical functions are limited to those approximating real optical elements for 3D display applications[17]. It is also difficult to fabricate large HOEs due to the difficulty of forming such a large wavefront in the recording process. Recently, to overcome the difficulty of large HOE fabrication, Bruder et al.[18] have reported a new HOE fabrication technique, in which the divided wavefront of the desired optical function is reconstructed using computer-generated holograms (CGHs) and then optically recorded in a tiling manner to form the entire size of a transparent-type HOE.

Here we report a projection-type holographic 3D display with a high degree of freedom of the display size and visual angle. Our display system is achieved by a combination of digital holographic projection and a digitally designed HOE (DDHOE). Because the DDHOE can have an arbitrary concave reflection function, an enlarged holographic image projected on the screen is concentrated into a target observation area with a large visual angle. Our approach can also resolve the above problems pointed by Komar[13], since the DDHOE does not require any large optical elements to form the wavefront of the optical function. Moreover, the DDHOE can have an appropriate reflection function to cancel out the distortion factor, and digital holographic projection can correct the mismatch of axial and lateral magnifications on the hologram data for distortion-free imaging. In comparison with Bruder's[18] technique, because our DDHOE is fabricated in the manner of reflection-type hologram recording, practically no effect on the light coming from behind the HOE can be realized with higher wavelength selectivity, which is an important factor for applications such as see-through 2D or 3D head-up displays and head-mounted displays.

## Results

**Spatial-temporal limitation of typical holographic displays.** Consider the simple one-dimensional case as an example (Fig. 1a), where the display width, $W$, is defined as $W = p \times M$, where $p$ is the pixel period and $M$ is the number of pixels in the display device. The maximum diffraction angle, $\theta_{\text{DIF}}$, is defined as $\theta_{\text{DIF}} = 2\sin^{-1}(\lambda/2p)$, where $\lambda$ is the wavelength. Under the condition that a plane wave illuminates the hologram displayed on an SLM, the minimum distance, $Z_{\text{MIN}}$, at which the entire display area can be viewed is given by $Z_{\text{MIN}} = W/(2\tan(\theta_{\text{DIF}}/2))$, with visual angle $\theta_{\text{VIS}}$ identical to the above maximum diffraction angle $\theta_{\text{DIF}}$ (Fig. 1a). In the SBP theory[10], a stationary observer is assumed to be in front of the holographic display, as shown in Fig. 1a. The SBP is a product of the hologram size, given as the 'space', $W$, and the maximum spatial frequency of the hologram, given as the 'bandwidth', $f_{\text{MAX}} = \sin(\theta_{\text{VIS}})/\lambda$. The bandwidth is not only related to the visual angle, but also defines the sharpness of the reconstructed contents. If $W$ is enlarged by increasing the pixel period, $\theta_{\text{DIF}}$ and $\theta_{\text{VIS}}$ become small, and vice versa. Thus, it is clear that the SBP cannot be increased as long as the display resolution $M$ is fixed. Even an 8 K ($7{,}680 \times 4{,}320$ pixels) display resolution cannot achieve a suitable display size with a visual angle large enough for practical use. In the case where an 8 K display device with a pixel period of $4.8\,\mu m$ is used, the above

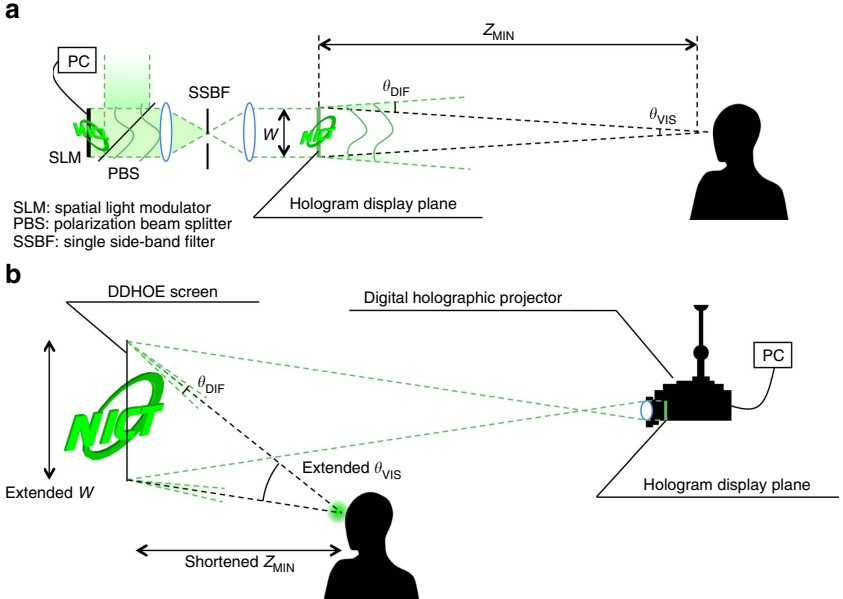

**Figure 1 | General electronic holography system and the new approach.** (**a**) $Z_{MIN}$ is the minimum observation distance for viewing the entire display area. There is a trade-off between the display size $W$ and the visual angle $\theta_{VIS}$ due to the limitation of the maximum diffraction angle $\theta_{DIF}$. (**b**) The new approach combining the digitally designed holographic optical element (DDHOE) screen and the holographic projector. Owing to the arbitrary reflection function of the DDHOE fabricated by a wavefront printing technique[20–23], the holographic image enlarged by projection is concentrated to a target observation area with high freedom in the design of $W$, $\theta_{VIS}$ and $Z_{MIN}$.

parameters in the horizontal direction are $W = 3.6\,\text{cm}$, $\theta_{DIF} = \theta_{VIS} = 6.8°$ and $Z_{MIN} = 32.4\,\text{cm}$ at $\lambda = 532\,\text{nm}$ (ref. 19). If $W$ is doubled to $7.2\,\text{cm}$ by increasing the pixel period to $P = 9.6\,\mu\text{m}$, $\theta_{DIF}$ and $\theta_{VIS}$ are roughly halved to $3.17°$ and $Z_{MIN}$ becomes $129.8\,\text{cm}$.

**Digital holographic projection incorporating DDHOE screen.** The proposed holographic 3D display is constructed with a digital holographic projection technique and a holographic screen of DDHOE (Fig. 1b). The DDHOE is a kind of HOE, but is fabricated by using a wavefront printing technique[20–22]. The wavefront printing technique has recently been proposed as a static hologram recording technique for high-definition 3D visualization, in which the wavefronts of 3D objects or scenes are reproduced by CGHs and then recorded as volume holograms. For 3D visualization, owing to the recent progress in high-speed computing techniques, our wavefront printer is capable of printing over 77 billion pixels of wavefront information in a $10 \times 10\,\text{cm}^2$ hologram (Fig. 2a,b)[23]. By replacing CGHs of 3D objects or scenes with those of the desired optical functions, this technique has the potential to offer reflection-type volume HOEs with digitally designed arbitrary optical characteristics. In the DDHOE fabrication process, since the entire hologram recording is divided into sub-regions and executed sequentially in a tiling manner, the HOE size is not limited. Unlike recent diffractive optical elements using CGH techniques and fabricated by milling or microlithography[24], our DDHOE has low weight, flexibility and wavelength selectivity owing to the use of the volume hologram recorded on a photopolymer holographic recording film, which is one of the requirements for realizing see-through screens for a wide range of applications.

**Design of DDHOE screen.** For our display system, we first designed the appropriate reflection function of the DDHOE that will concentrate the large holographic image projected by the holographic projector at the target observation point (Fig. 3a).

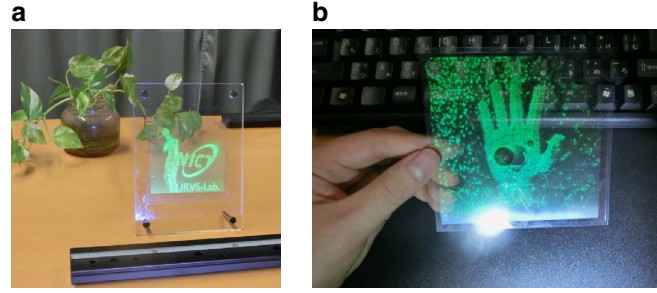

**Figure 2 | Some examples of our wavefront printing technique.** (**a,b**) $10 \times 10\,\text{cm}^2$ of 3D object data were printed out on transparent sheets[23].

The modulation angle, $\theta_M$, in the reflection of the incident light was derived from two factors: the specular direction of the incident light from the centre of the holographic projection, $\theta_S$, and the direction of the assumed observation point, $\theta_O$, from the current sampling point on the DDHOE (Fig. 3a). We set the centre of the projection at $(x, y, z) = (0, 0, 100\,\text{cm})$ and the observation point at $(x, y, z) = (0, 5.9, 20\,\text{cm})$ in the $XYZ$ coordinate system depicted in Fig. 3a, in which the origin was set at the centre of the holographic screen. $\theta_M$ was then translated into the phase distribution, $\varphi$, and it was encoded into the hologram data that will reproduce the wavefront of $\varphi$ as the object light in the hologram recording process. In the wavefront printing, the entire hologram data of the DDHOE was divided and each sub-hologram data reconstructed part of the object light at $532\,\text{nm}$ of wavelength in the same manner as an electronic holography system[19]. The reconstructed each object light was optically recorded as a reflection-type volume hologram in a tiling manner by using motorized $X$–$Y$ stages to fabricate the entire DDHOE. Figure 4a shows the optical behaviour of the fabricated holographic screen demonstrated in an artificial fog. The reflected light on the screen was clearly concentrated at the target observation point. Note that although our demonstration of the

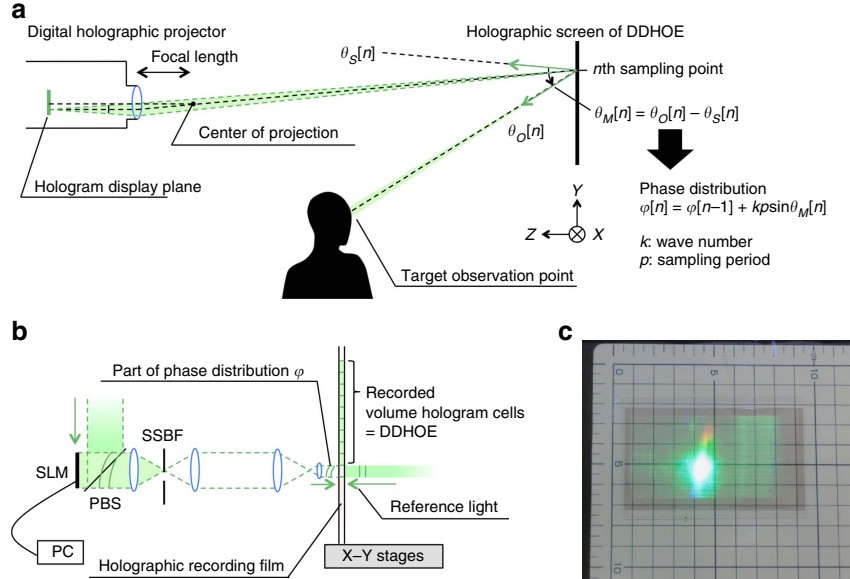

**Figure 3 | Design and fabrication of digitally designed holographic optical element.** (**a**) The reflection function of the digitally designed holographic optical element (DDHOE) is derived as a discretized phase distribution $\varphi[n]$ from two factors: the direction to the target observation area $\theta_O[n]$ and the specular direction $\theta_S[n]$ from the centre of the projection. (**b**) The object light of the wavefront $\varphi$ is divided and sequentially recorded on a photopolymer holographic recording film as a volume hologram. (**c**) The screen of the DDHOE with a size of $73.6 \times 41.4 \, \text{mm}^2$ and a designed reflection function was fabricated for the experiment in this paper.

DDHOE was in monochromatic green, the fabrication of multicolour HOEs has been reported[25], and thus, it will be possible to fabricate a full-colour DDHOE for full-colour holographic reconstruction in the near future.

**Digital holographic projection.** A holographic projector was prepared by adapting an electronic holography system[19], in which an 8 K (7,680 × 4,320 pixels) liquid crystal on silicon (LCOS)-SLM with a pixel period of 4.8 µm is used for optical reconstruction of the hologram data. The projection magnification was set at 2.0 and, therefore, the size of the projected wavefront on the holographic screen was $73.6 \times 41.4 \, \text{mm}^2$.

**Holographic reconstruction.** For our display, $\theta_{\text{VIS}}$ was increased to 20.8° and $Z_{\text{MIN}}$ was shortened to 20 cm at the target observation point via the holographic screen of the DDHOE, compared with the corresponding values of 3.17° and 129.8 cm for a general holographic display[11], with the same pixel number and display size. Figure 4b,c shows an optical demonstration of the effect of the DDHOE compared with the case of using a half mirror as a screen. A holographic image of a checkered pattern located on the screen and having the same size as the screen was observed by a camera. It is clear that the whole-projected area on the DDHOE was observed successfully due to the effect of the appropriate concave reflection function, while the observable display area was greatly limited when using the half mirror. Figure 4e shows the optical reconstruction results of a 3D scene, in which two checkered square objects located 1 and 5 cm behind the holographic screen and focused on in Fig. 4d,f were optically reconstructed. These results showed that the optical characteristics of the DDHOE are theoretically good enough to increase both the display size and visual angle of a projection-type holographic 3D display.

**Discussion**

Our display system, namely, holographic projection with a DDHOE as a holographic screen, is one of the improved solutions to overcome the restriction of the pixel resolution of current display devices even though the observable area will be limited. The proposed system has achieved the twofold enlargement of the display size and the sixfold enlargement of the visual angle compared with a conventional holographic display[19].

Note that our display does not increase the maximum diffraction angle, that is, the observable area of a holographic image is still limited by the spatio-temporal resolution of current display devices. However, the high degree of freedom of the display size and visual angle of our display system will allow limited, but new practical applications of holographic 3D displays, such as in-car head-up displays, smart-glasses and head-mounted displays; they will permit the design of displays with a narrow observable area. In addition, the fabrication of a full-colour display system with a higher diffraction efficiency of the DDHOE will be considered in future work to more clearly show the prospect of our approach for practical use, since several papers[25,26] have shown the potential of the fabrication of full-colour and brighter HOEs. Also, multiple holographic projections from different incident directions to the DDHOE will enable us to extend the observable area.

Since the DDHOE has several advantages compared with the general diffractive optical element/HOEs, such as the freedom of applying optical functions without any real optical elements to the object light of an arbitrary wavefront in fabrication, the ease of fabricating large screens and good transparency due to its wavelength selectivity, the applications of the DDHOE are not limited to only holographic 3D displays, but also cover other fields, such as wearable displays, the beam shaping of laser light sources, the formation of reference wavefronts for aspheric and freeform elements in interferometry, and so forth.

**Methods**

**Geometrical design of DDHOE screen.** In the fabrication of the reflection-type DDHOE, the reflection function of the DDHOE was appropriately designed by considering the parameters of the optical set-up for reconstruction, that is, the target observation point $O = (x_O, y_O, z_O)$ and the centre of the holographic projection $P = (x_P, y_P, z_P)$, as shown in Fig. 5.

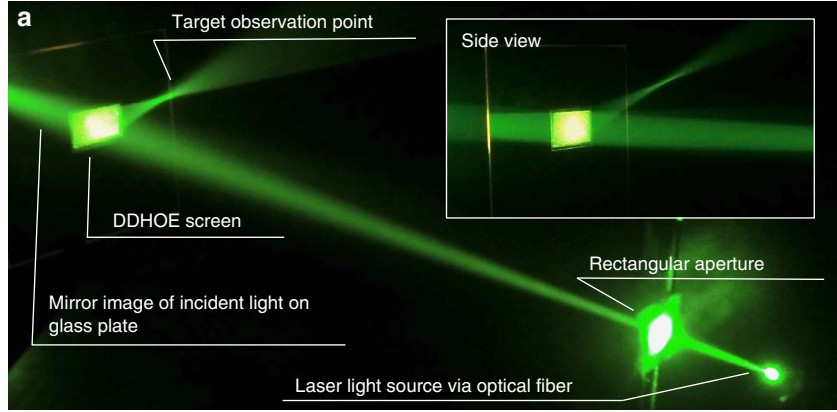

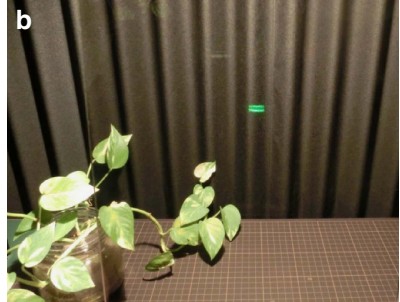

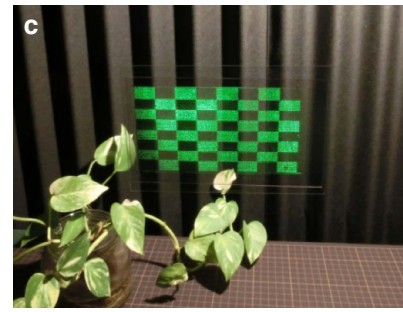

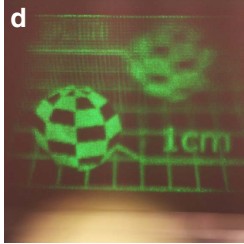
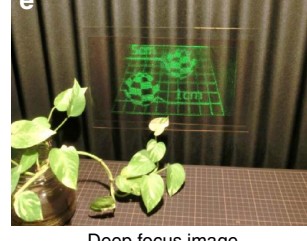
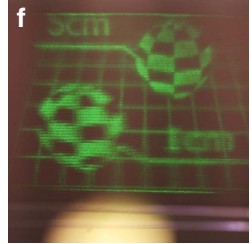

Focusing on 1 cm | Deep focus image | Focusing on 5 cm

**Figure 4 | Experimental results.** (**a**) Demonstration of the optical behaviour of the fabricated digitally designed holographic optical element (DDHOE) in artificial fog. The diffraction efficiency of the DDHOE was 52.9%. (**b,c**) Comparison between the half-mirror case (**b**) and the holographic screen of the DDHOE (**c**) by reconstruction of the checkered board observed at the target observation point. The backgrounds were the plant and the accordion screen. (**d–f**) Reconstruction of three-dimensional scene with two checkered squares 1 and 5 cm behind the holographic screen.

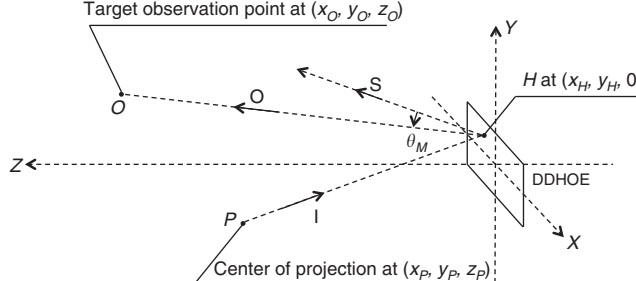

**Figure 5 | Reconstruction geometry for designing the function of the digitally designed holographic optical element.** Two paths; the specular direction **S** of the incident light from $P$ via the digitally designed holographic optical element (DDHOE) and the direction **O** to the target observation point at $H$, determine the modulation angle $\theta_M$ for the $x$ and $y$ directions to be implemented on the DDHOE.

The direction from $P$ to the point $H = (x_H, y_H, 0)$ on the DDHOE is indicated by **I** in vector form as

$$\mathbf{I} = l_I\mathbf{i} + m_I\mathbf{j} + n_I\mathbf{k}, \tag{1}$$

where **i**, **j** and **k** are the unit vectors in the $x$, $y$ and $z$ directions, respectively. The unnormalized direction cosines of **I** can be written in terms of the coordinates of $H$ and $P$ as

$$L_I = X_H - X_P, \tag{2}$$

$$M_I = Y_H - Y_P, \tag{3}$$

$$N_I = -Z_P. \tag{4}$$

Moreover, the specular direction of $I$ at $H$ indicated by **S** and its unnormalized direction cosines are written as

$$\mathbf{S} = l_S\mathbf{i} + m_S\mathbf{j} + n_S\mathbf{k}, \tag{5}$$

$$l_S = X_H - X_P, \tag{6}$$

$$m_S = Y_H - Y_P, \tag{7}$$

$$n_S = Z_P. \tag{8}$$

In the same manner, the vector **O** is defined as the direction $H$ to $O$ and can be written as

$$\mathbf{O} = l_O\mathbf{i} + m_O\mathbf{j} + n_O\mathbf{k}, \tag{9}$$

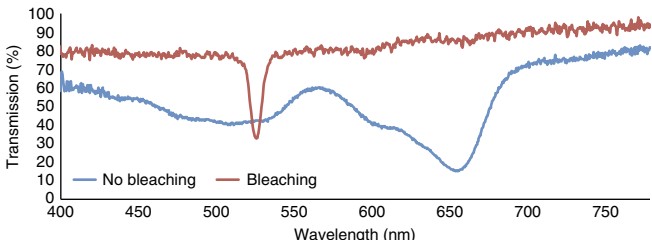

**Figure 6 | Transmission curves of the digitally designed holographic optical element.** After the bleaching process, the digitally designed holographic optical element (DDHOE) achieved high transmission characteristics at visible light wavelengths. Note that the transmission around 532 nm of wavelength is low due to the reflection function of DDHOE to incident light.

where the unnormalized direction cosines of **O** are

$$l_O = X_O - X_H, \tag{10}$$

$$m_O = Y_O - Y_H, \tag{11}$$

$$n_O = Z_O. \tag{12}$$

If the reflection-type DDHOE has no directional modulation of the incident light, that is, it behaves as a normal mirror, the incident light will be reflected divergently to **S** and thus the observer cannot observe the entire holographic image, since only a small part of the reflection light will reach the observer's pupil. The purpose of using the DDHOE as a holographic screen is to modulate the reflection direction of the projected holographic image to the target observation point O. The modulation angles $\theta_M$ from the specular direction **S** at H for the x and y directions are written as

$$\theta_M(X_H, Y_H) = (\theta_X(X_H), \theta_Y(Y_H)), \tag{13}$$

where

$$\theta_X(X_H) = \arctan(l_O/n_O) - \arctan(l_S/n_S), \tag{14}$$

$$\theta_Y(Y_H) = \arctan(m_O/n_O) - \arctan(m_S/n_S). \tag{15}$$

By modulating the reflection angle in accordance with $\theta_X$ and $\theta_Y$, the light of the projected hologram image will be accumurated at O. Here, for simplicity, we considered the Y–Z coordinates only, and the DDHOE was discretized with a sampling period p. Since the incident direction of the reference wave in the recording process was set to be normal to the recording material surface, the propagation direction of the object wave at np, the coordinate of the n th sampling point, should become $\theta_Y(np)$. For the CGH calculation step, $\theta_Y(np)$ is transformed into a discretized phase distribution φ[n] using a recurrence formula from the neighbouring sampling point, $\varphi[n] = \varphi[n-1] + kp\sin\theta_Y[n]$, where k is the wave number. This product φ[n] was then multiplied by a uniform amplitude distribution and encoded as a fringe pattern by calculating a real number at each sample. Note that, to eliminate undesirable light, a half-zone plate process was used[27]; that is, the propagation direction $\theta_Y[n]$ was restricted to half in the vertical direction in the case of our wavefront printer. The whole fringe pattern was divided into a set of sub-holograms for display on the SLM in the wavefront printing step (Fig. 3b). The wavefront propagation of the overall display system from the holographic projector to the observable plane via the DDHOE screen is mathematically described on the basis of wave optics in Supplementary Note 1 and Supplementary Fig. 1.

**Fabrication of DDHOE screen.** The wavefront printer has a 4 K (4,320 × 2,160 pixels) SLM panel with a pixel period of 3.5 μm, and the formed sub-holograms of 3,600 × 1,800 pixels were sequentially displayed on the SLM and optically reconstructed by a collimated plane wave at a wavelength of 532 nm. After passing through a single-sideband filter[27], the reduction optical system demagnified the reproduced wavefront by a factor of 0.102, and the wavefront was then recorded on a holographic recording film (Covestro Bayfol HX-102 photopolymer material) by interference with reference light. The holographic recording film after the recording of all sub-holograms was then processed by bleaching for high transmission of visible light. The transmission curves of our DDHOE before and after the bleaching process were measured by using a B&W TEK BTC112E spectrometer (Fig. 6). The diffraction efficiency of the fabricated DDHOE was measured to be 52.9%, which is not a maximized value due to a lack of optimization of the exposure energy and the ratio of the object/reference light in the recording process. By optimizing these recording factors, ideally the diffraction efficiency can reach over 90% (ref. 26) using the same holographic recording film, and so maximization of the diffraction efficiency of the DDHOE will be performed in our future work. The main parameters of our experiments, the fabrication of the DDHOE and the holographic projection, are listed in Table 1.

**Table 1 | Main parameters.**

| *Fabrication of DDHOE* | |
|---|---|
| Wavelength of laser | 532 nm |
| Size of DDHOE | 73.6 × 41.4 mm |
| Size of hologram data | 204,800 × 115,200 pixel |
| Sampling period p of DDHOE | 0.36 μm |
| Size of sub-hologram | 3,600 × 1,800 pixel |
| Hologram recording film | Bayfol HX-102 by Covestro |
| | |
| *Holographic projector* | |
| Wavelength of laser | 532 nm |
| Size of SLM | 7,680 × 4,320 pixel |
| Pixel period of SLM | 4.8 μm |
| Magnification of projection | 2.0 |
| Focal length of projection lens | 500 mm |

**Calculation of hologram data.** For the reconstruction of 3D scenes and objects, the hologram data on the holographic projector must be calculated by considering the distortion caused by the reflection function of the holographic screen. One possible calculation method involves the deformation of 3D space composed of point/polygon light sources based on the inverse distortion function of a DDHOE before the propagation calculation from the light sources to the hologram plane. Another way, employed in this article, is to correct the light ray directions via a HS-based calculation[28]. In the HS-based calculation, dense rays sampled on the hologram plane can be obtained as a set of perspective images taken by a virtual camera array. By correcting each camera rotation by considering the reflection direction at the DDHOE, hologram data without any distortion caused by the DDHOE could be generated.

**Data availability.** The data that support the findings of this study are available from the corresponding author upon request.

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

## Acknowledgements

This work was supported by JSPS KAKENHI grant number 26790064 and 16H01742. This research and development work was supported by the MIC/SCOPE # 162103005. This research is partially supported by the Centre of Innovation Program from Japan Science and Technology Agency, JST. We thank K. Terada for the support in making the artificial fog in Figure 4a. We also thank R. Usuha for generating the 3D object data used in Figure 2b.

## Author contributions

K.W. constructed the holographic screen of the DDHOE, performed the optical experiments and wrote the article. P-Y.H. designed the holographic screen of the DDHOE. T.S. designed, developed and aligned the optical system of the holographic projection system. R.O. established the environment for exposure/post treatment of the photopolymer film and discussed the specifications of the DDHOE. H.S., Y.I., M.O and Y.-P.H. discussed the results and commented on the manuscript. K.Y. did project planning and management. All authors reviewed the manuscript.

## Additional information

**Competing financial interests:** Four patents relevant to this work are pending in Japan. The patent numbers are 2015-095297, 2015-095298, 2015–209713 and 2015–210488.

