## [Peer Review File · Nature Communications]

Reviewers' Comments:

Reviewer #1 (Remarks to the Author):

The authors describe the use of a computed reflective holographic optical element as the screen in a holographic system. The paper is clearly written and well organized but some changes are needed before it should be considered for publication.

While the system described is somewhat novel and potentially of interest to practitioners in the field of holographic displays, it is not as novel as the authors suggest. A similar arrangement (though using an "analog" HOE screen, and arranged for multiple view zones) was used by Komar in his holographic cinema system -- see for example Victor G. Komar; "Principle Of The Holographic Cinematography". Proc. SPIE 0136, 1st European Conf on Optics Applied to Metrology, 358 (April 18, 1978); doi:10.1117/12.956185.

Also, discussions of the tradeoffs possible between the size and view angle of the reconstructed lightfield from a hologram are conventionally formulated in terms of space-bandwidth product. See for example L. Onural, F. Yaras and H. Kang, "Digital Holographic Three-Dimensional Video Displays," in Proceedings of the IEEE, vol. 99, no. 4, pp. 576-589, April 2011. doi: 10.1109/JPROC.2010.2098430

A minor omission is that in the discussion of the 8K display on p3, the authors do not specify what value they are using for the illumination wavelength.

This reviewer requests that the authors discuss their contribution in the context of Komar's earlier work, and also that they make a somewhat more rigorous space-bandwidth product discussion of the design considerations for the display system.

Reviewer #2 (Remarks to the Author):

The authors report a digitally designed holographic optical element which is then used as part of a holographic display. The designed element is then manufactured and tested and some results are presented. The designed element is a large array of pixels; seemingly, an arbitrary wavefront function is sequentially printed on the element. In the presented case, as I understand, the device acts effectively as an aspherical optical element which is capable of focusing an incident collimated beam to a designated point at a specific location. As a consequence of that focusing ability, a larger viewing angle is achieved. The original holographic pattern projected out by a projector, its wavefront function which encodes the 3D information as holographic fringes are then modified by the designed optical element; a 3D reconstruction is observed by a viewer at the designated location.

Here are the key achievements as reported by the paper:

- * The designed large optical element is a novelty
- * The designed element certainly serves the purpose as demonstrated by the figures

Here are some points which might be further improved:

- * Though it is not necessary for this rather experimental work report, a mathematical description of the entire optical path would be valuable to eliminate potential ambiguities.

* Authors use the term "pitch" to describe the pixel period; this is quite common in the community, and therefore, acceptable, and will still be clearly understood by the readers. However, as described in the dictionaries, literally, the word "pitch" refers to frequency, which is the reciprocal of period.

List of references is adequate. The presentation is clear and appropriate.

I recommend the publication of this paper.

Reviewer #3 (Remarks to the Author):

This manuscript discusses the creation of computer generated HOEs for use in expanding the field of view and display size of a holographic projection system. It also displays preliminary results of the integration of the HOE and the projection system.

This type of "DDHOE" is not novel. A quick search turned up a number of publications. See for example "Diffractive optics in large sizes: computer-generated holograms (CGH) based on Bayfol® HX photopolymer" doi: 10.1117/12.2077139. The projector itself has been previously published as well. The novelty in this case comes from replacing a physical aspheric concave mirror surface with a volume hologram, but the hologram has not been optimized (a standard procedure in any new printing process). 52.9% DE in single color is not reasonable. The above mentioned reference achieved 88% DE in a similar application.

The manuscript ends quite abruptly - is there a last page or conclusion section that was missing in the submission?

I suggest that for a significant technological advancement to be present in this work, higher diffraction efficiency and full color display should be exhibited. As these are the next steps listed in the manuscript, I suggest re-submission after the completion of this work.

RESPONSES TO REVIEWERS

We wish to express our appreciation to the editor and reviewers for their insightful comments regarding our manuscript. We have modified the paper in accordance with your comments and added some new descriptions. The details of these changes and the responses to the reviewers' comments are described below.

Reviewer #1 comment: -----

"While the system described is somewhat novel and potentially of interest to practitioners in the field of holographic displays, it is not as novel as the authors suggest. A similar arrangement (though using an "analog" HOE screen, and arranged for multiple view zones) was used by Komar in his holographic cinema system -- see for example Victor G. Komar; "Principle Of The Holographic Cinematography". Proc. SPIE 0136, 1st European Conf on Optics Applied to Metrology, 358 (April 18, 1978); doi:10.1117/12.956185."

Authors' Response:

Thank you for this valuable point. Actually, Komar reported the primal concept of the system composed of a holographic screen and a holographic projection. However, since his holographic screen and the holographic film for the projection were both analog holograms, Komar himself pointed out several problems of his demonstration, namely, the difficulty of fabricating a large holographic screen, the mismatch of axial and lateral magnifications in the projection, and also the image distortion due to observing from an oblique direction.

Our approach can solve all these problems since the DDHOE does not require any large optical elements to form the wavefront of the optical function; thus, the fabrication size is not limited. Moreover, the DDHOE can have an optimized aspheric function to cancel out the distortion factors. Digital holographic projection enables us to correct the mismatch of the axial and lateral magnifications by deforming the object shape in the hologram calculation step for distortion-free imaging.

We consider that our demonstration has shown the real potential of Komar's idea for the first time by using "digital" technology to solve the critical problems and that it satisfies the criterion of novelty for publication in Nature Communication.

We have added this discussion in the Background section on pages 4 and 5.

Reviewer #1 comment: -----

"Also, discussions of the tradeoffs possible between the size and view angle of the reconstructed lightfield from a hologram are conventionally formulated in terms of

space-bandwidth product. See for example L. Onural, F. Yaras and H. Kang, "Digital Holographic Three-Dimensional Video Displays," in Proceedings of the IEEE, vol. 99, no. 4, pp. 576-589, April 2011. doi: 10.1109/JPROC.2010.2098430."

Authors' Response:

We are really grateful for your helpful advice. As the reviewer pointed out, the SBP is a good formula for explaining the trade-off between the size and the visual angle (=viewing angle) of the conventional display. We have modified the Background section to introduce the theory of the SBP.

Reviewer #1 comment: -----

"A minor omission is that in the discussion of the 8K display on p3, the authors do not specify what value they are using for the illumination wavelength."

Authors' Response:

We have added the wavelength on p. 3 in accordance with the reviewer's comment.

Reviewer #2 comment: -----

"* Though it is not necessary for this rather experimental work report, a mathematical description of the entire optical path would be valuable to eliminate potential ambiguities."

Authors' Response:

We thank the reviewer for the insightful suggestion. In the Methods section, we have added a description of the relationship between the optical paths from the center of the holographic projection to the target observation point via the reflection-type holographic screen of the DDHOE based on the vector form.

Reviewer #2 comment: -----

"Authors use the term "pitch" to describe the pixel period; this is quite common in the community, and therefore, acceptable, and will still be clearly understood by the readers. However, as described in the dictionaries, literally, the word "pitch" refers to frequency, which is the reciprocal of period."

Authors' Response:

The reviewer is correct, and to make the manuscript more readable, we have replaced “pixel pitch” with “pixel period” and “sampling pitch” with “sampling period” throughout the manuscript.

Reviewer #3 comment: -----

“This type of "DDHOE" is not novel. A quick search turned up a number of publications. See for example "Diffractive optics in large sizes: computer-generated holograms (CGH) based on Bayfol®; HX photopolymer" doi:10.1117/12.2077139. The projector itself has been previously published as well. The novelty in this case comes from replacing a physical aspheric concave mirror surface with a volume hologram, but the hologram has not been optimized (a standard procedure in any new printing process). 52.9% DE in single color is not reasonable. The above mentioned reference achieved 88% DE in a similar application.”

And

“I suggest that for a significant technological advancement to be present in this work, higher diffraction efficiency and full color display should be exhibited. As these are the next steps listed in the manuscript, I suggest re-submission after the completion of this work.”

Authors’ Response:

Thank you for the comment. We consider that the main point of our manuscript is the novelty of combining the DDHOE screen with a holographic projection technique for the first time, and not the quality of the DDHOE screen.

As the reviewer pointed out, in the paper of Bruder et al., a similar HOE fabrication technique was demonstrated, in which a simple concave lens function was recorded onto a transparent-type HOE of 3 x 3 cm² hologram size in monochrome. However, they did not mention the application of HOEs as a special screen for the holographic projection. We fabricated the DDHOE by applying reflection-type holographic recording for projection screen use to obtain higher wavelength selectivity for the “see-through” 3D display system than that in their demonstration, which is an important factor for a wide range of applications such as in-car head-up displays, smart glasses, and head-mounted displays. We have added this discussion in the Background section on page 5.

Regarding making a full-color HOE and achieving high diffraction efficiency, we agree that it would be better if our display system realized full color with high diffraction efficiency. However, in a previous article on a holographic display system published in Nature [1], although the demonstrated holographic display system did not realize full color and the article did not mention the diffraction efficiency, it was published because they first demonstrated a display system in which a photorefractive polymer was used as the holographic recording material,

even though the photorefractive effect did not have any novelty in the optics field. We consider that the demonstration reported in our manuscript is also sufficient to show the novelty of our display system, even though it is a monochrome system with diffraction efficiency of 54%. Also, realizing a full-color system with high diffraction efficiency will take months, and the opportunity for the rapid dissemination of our technique will be lost. Therefore, these topics will be considered in our future work, as mentioned in the Result section on page 6 and the Discussion section on page 9.

[1] Blanche, P.-A. et al. Holographic three-dimensional telepresence using large-area photorefractive polymer *Nature* **468** (2010).

Reviewer #3 comment: -----

“The manuscript ends quite abruptly - is there a last page or conclusion section that was missing in the submission?”

Authors’ Response:

We are sorry for giving this impression when you read our manuscript. However, according to the guidelines of this journal, the Methods section should be placed at the end following the Discussion section, and there is no conclusion section as in other journals. The explanation concerning this point can be found on the Web page

http://www.nature.com/ncomms/authors/content_types.html, as follows:

“The main text of an Article should begin with an introduction (without heading) of referenced text that expands on the background of the work (some overlap with the abstract is acceptable), followed by sections headed Results, Discussion (if appropriate) and Methods (if appropriate). ”

RESPONSES TO REVIEWERS

We sincerely thank the editor and reviewers for their second review of our manuscript. We have corrected the paper in accordance with the remaining concerns of Reviewer #2. The details of these changes and the responses to Reviewer #2 are shown below.

Reviewer #2 comment: -----

"1- As a response to one of my earlier comments which says the paper lacks mathematical rigor to explain the wave propagation along the optical path, the authors now added some basic vector descriptions on pages 9 and 10. Their new added analysis is trivial, and therefore, not needed at all; however, such new content does not create any harm, either. When I said "...lacking mathematical description." in my earlier comment, what I meant was a clear and complete mathematical description of the wavefront propagation along the optical path. Such a content is still completely missing. However, again as I indicated in my first review, this is an optional recommendation, and therefore, should not be a basis to reject the paper for publication. Such a rigorous mathematical description of wave propagation, especially through complicated optical elements is a difficult task, anyway."

Authors' Response:

We are grateful to the reviewer for alerting us to the lack of a mathematical description of wave propagation. As the reviewer commented, this discussion may be considered as being supplemental and thus we have added a supplementary note for this purpose. In the supplementary note, the wave propagations from the holographic projector to the target observable point via the DDHOE screen are mathematically described step by step. We hope that the new description addresses your point.

Reviewer #2 comment: -----

"2- The word "optimized" and "optimal" appears now in the newly added content. I believe authors do not use this word correctly (this is unfortunately quite common in the scientific community): I believe what they try to say is "improved", instead. Clearly, when something is optimal, within the given descriptions and constraints, it must really be the best and cannot be further improved by anyone at any future time."

Authors' Response:

We completely agree with the reviewer's point about "optimized" and "optimal". Thank you for suggesting the word "improved". We have replaced "optimized solution" with "improved solution" in Discussion, and in other sections, we have replaced "optimized" with "appropriate" or "appropriately" as follows:

Moreover, the DDHOE can have an **appropriate** reflection function to cancel out the distortion factor, and digital holographic projection can correct the mismatch of axial and lateral magnifications on the hologram data for distortion-free imaging.

in INTRODUCTION.

Design of DDHOE screen

For our display system, we first designed the **appropriate** reflection function of the DDHOE that will concentrate the large holographic image projected by the holographic projector at the target observation point (Fig. 3a).

in RESULTS.

DISCUSSION

Our display system, namely, holographic projection with a DDHOE as a holographic screen, is one of the **improved** solutions to overcome the restriction of the pixel resolution of current display devices even though the observable area will be limited.

in DISCUSSION.

Geometrical design of DDHOE screen

In the fabrication of the reflection-type DDHOE, the reflection function of the DDHOE was **appropriately** designed by considering the parameters of the optical setup for reconstruction, i.e., the target observation point $O = (x_o, y_o, z_o)$ and the center of the holographic projection $P = (x_p, y_p, z_p)$, as shown in Fig. 5.

in METHODS.

Reviewers' Comments:

Reviewer #1 (Remarks to the Author):

I thank the authors for their attention to my previous comments. I think they have satisfactorily addressed them in the revised manuscript.

I have no objection to publication of this revised manuscript.

Reviewer #2 (Remarks to the Author):

I carefully studied the comments of other reviewers, and the revised version of the paper.

I still keep my earlier position during the first round of reviews: authors present a novel holographic display by incorporating a digital HOE.

I also see that the manuscript is somewhat further improved by taking the comments into consideration. However, there are now two minor issues that seem to be somewhat degrading the paper:

1- As a response to one of my earlier comments which says the paper lacks mathematical rigor to explain the wave propagation along the optical path, the authors now added some basic vector descriptions on pages 9 and 10. Their new added analysis is trivial, and therefore, not needed at all; however, such new content does not create any harm, either. When I said "...lacking mathematical description." in my earlier comment, what I meant was a clear and complete mathematical description of the wavefront propagation along the optical path. Such a content is still completely missing. However, again as I indicated in my first review, this is an optional recommendation, and therefore, should not be a basis to reject the paper for publication. Such a rigorous mathematical description of wave propagation, especially through complicated optical elements is a difficult task, anyway.

2- The word "optimized" and "optimal" appears now in the newly added content. I believe authors do not use this word correctly (this is unfortunately quite common in the scientific community): I believe what they try to say is "improved", instead. Clearly, when something is optimal, within the given descriptions and constraints, it must really be the best and cannot be further improved by anyone at any future time.

Based on these observations, I am in favor of publication of this paper, maybe with a few further minor modifications.

RESPONSES TO REVIEWERS

We sincerely thank the editor and reviewers for their second review of our manuscript. We have corrected the paper in accordance with the remaining concerns of Reviewer #2. The details of these changes and the responses to Reviewer #2 are shown below.

Reviewer #2 comment: -----

"1- As a response to one of my earlier comments which says the paper lacks mathematical rigor to explain the wave propagation along the optical path, the authors now added some basic vector descriptions on pages 9 and 10. Their new added analysis is trivial, and therefore, not needed at all; however, such new content does not create any harm, either. When I said "...lacking mathematical description." in my earlier comment, what I meant was a clear and complete mathematical description of the wavefront propagation along the optical path. Such a content is still completely missing. However, again as I indicated in my first review, this is an optional recommendation, and therefore, should not be a basis to reject the paper for publication. Such a rigorous mathematical description of wave propagation, especially through complicated optical elements is a difficult task, anyway."

Authors' Response:

We are grateful to the reviewer for alerting us to the lack of a mathematical description of wave propagation. As the reviewer commented, this discussion may be considered as being supplemental and thus we have added a supplementary note for this purpose. In the supplementary note, the wave propagations from the holographic projector to the target observable point via the DDHOE screen are mathematically described step by step. We hope that the new description addresses your point.

Reviewer #2 comment: -----

"2- The word "optimized" and "optimal" appears now in the newly added content. I believe authors do not use this word correctly (this is unfortunately quite common in the scientific community): I believe what they try to say is "improved", instead. Clearly, when something is optimal, within the given descriptions and constraints, it must really be the best and cannot be further improved by anyone at any future time."

Authors' Response:

We completely agree with the reviewer's point about "optimized" and "optimal". Thank you for suggesting the word "improved". We have replaced "optimized solution" with "improved solution" in Discussion, and in other sections, we have replaced "optimized" with "appropriate" or "appropriately" as follows:

Moreover, the DDHOE can have an **appropriate** reflection function to cancel out the distortion factor, and digital holographic projection can correct the mismatch of axial and lateral magnifications on the hologram data for distortion-free imaging.

in INTRODUCTION.

Design of DDHOE screen

For our display system, we first designed the **appropriate** reflection function of the DDHOE that will concentrate the large holographic image projected by the holographic projector at the target observation point (Fig. 3a).

in RESULTS.

DISCUSSION

Our display system, namely, holographic projection with a DDHOE as a holographic screen, is one of the **improved** solutions to overcome the restriction of the pixel resolution of current display devices even though the observable area will be limited.

in DISCUSSION.

Geometrical design of DDHOE screen

In the fabrication of the reflection-type DDHOE, the reflection function of the DDHOE was **appropriately** designed by considering the parameters of the optical setup for reconstruction, i.e., the target observation point $O = (x_o, y_o, z_o)$ and the center of the holographic projection $P = (x_p, y_p, z_p)$, as shown in Fig. 5.

in METHODS.